# Evaluating the Renewal Degree for Expressway Regeneration Projects Based on a Model Integrating the Fuzzy Delphi Method, the Fuzzy AHP Method, and the TOPSIS Method

**Min Zhu** [1,2,3], **Wenbo Zhou** [1,2,*], **Min Hu** [1,2], **Juan Du** [1,2] and **Tengfei Yuan** [1,2]

1    SHU-UTS SILC Business School, Shanghai University, Shanghai 201800, China
2    Shanghai Engineering Research Center of Urban Infrastructure Renewal, Shanghai 200032, China
3    School of Engineering and Management, Pingxiang University, Pingxiang 337000, China
*    Correspondence: zhouwb@stec.net

**Abstract:** As the volume and scale of urban expressways continue to increase, renewal remains a concern for urban development. The renewal and decision-making of an urban expressway need to be endowed with new concepts to adapt to the rapid development of cities. Nevertheless, in addition to considering road factors such as facility conditions, driving conditions, and environmental protection, the existing evaluation system lacks comprehensive consideration of factors that improve resilience and adapt to future urban development, and it lacks a quantifiable general update evaluation system. Thus, the establishment of a comprehensive renewal indicator system and a mixed evaluation framework is a challenge. This study proposes an evaluation framework of expressway renewal indicators that integrates the three dimensions of macro, meso, and micro based on the fuzzy Delphi method, the fuzzy AHP method, and the TOPSIS method. A q-rung orthopair fuzzy linguistic set was used to handle expert uncertainty information in the process of conducting fuzzy evaluations. The indicators were refined into general and quantifiable evaluation indicators to improve their versatility. Moreover, the renewal value of expressways was measured and calculated using the TOPSIS method, and four renewal intervals were divided according to the calculation results. As a result, 28 renewal indicators were screened out, and the five factors with the greatest impact on renewal were the demand for transport development, the renewal of facility and service functions, the upgrading of institutional resilience, structural renewal, and economic development. The model was applied to eight expressways in Shanghai to calculate the renewal degree value and divide the renewal status. The model could identify the renewal needs of each road to guide the renewal decision. This study proposes an evaluation model to measure urban expressway renewal studies and provides a reference for urban renewal in the area of sustainable development

**Keywords:** expressway renewal; renewal degree; fuzzy delphi method; fuzzy AHP; q-Rung orthopair fuzzy sets; TOPSIS

## 1. Introduction

Over the long history of urban development, large cities have been confronted with regeneration issues to varying degrees. Urban road regeneration, also known as urban road reconstruction and road expansion, is a part of infrastructure renewal under the concept of urban regeneration [1]. Urban expressways play a vital role in connecting and driving the development of cities as transport hubs and backbones, and their regeneration has received increasing attention [2]. Major cities around the world have high-density urban expressway networks [3]. Take the first-tier city of Shanghai, China, as an example. Its investment in transport facilities amounted to USD 142.13 billion between 2003 and 2017 [4], and relevant industry data show that the total length of Shanghai's road network reached 18,500 km in 2022, with an average operating life of 12.5 years [5]. Thus, for these roads, the problems associated with a long operating period must be dealt with. Moreover, the problems left

behind by the early high expressway planning restrict the development of the city, such as the relatively static high expressway planning, the lack of adaptation of the dynamic development of urban traffic, the ill-considered planning function, the uncoordinated planning road scale, and the regional industrial relationship [6]. Urban expressway renewal is one of the main ways to make up for such planning drawbacks, enhance functions and services, improve environmental impacts, and increase the convenience of travel for citizens. It is also an important guarantee to promote coordinated and sustainable development between cities [7], but how to carry out orderly expressway renewal each year amid financial constraints is a problem that needs to be studied.

Faced with the problem of building and renewing urban infrastructure, countries around the world have proposed relevant evaluation criteria to promote the green construction and sustainability of roads by way of assessment. BREEAM Infrastructure is a sustainability assessment, rating, and award scheme widely used in the UK and Ireland for civil engineering, infrastructure, landscaping, and public sector projects, whose main function is to verify the sustainability performance of projects, driving sustainability across the sector in the region [8]. The US EnvisionV3 system helps infrastructure stakeholders implement more sustainable, resilient, and equitable projects that reduce greenhouse gas emissions, create well-paying "green" jobs, address environmental justice issues, and achieve climate change adaptation goals [9]. These two evaluation systems mainly consider the three main elements of sustainable urban infrastructure development at a macro level but lack a targeted evaluation of road projects. BE2ST-in-Highways provides a quantitative analysis and rating system for sustainable highway construction in the United States for the planning and design of highway construction projects [10]. The Greenroads system, also from the United States, considers both the needs of the project and the impact of surrounding public facilities, allowing for a full life cycle assessment of the road [11]. The GreenPave system used in Canada is designed for pavement projects, with a particular focus on environmental engineering improvements [12]. In China, national or locally published standards and specifications are used for various technical evaluations during the construction phase of expressways [13]. Although these assessment systems focus on the road and the environmental impact, the evaluation level is relatively homogeneous. To fully reflect the functionality of urban expressways, a comprehensive and multidimensional evaluation system is needed to assist in renewal decisions.

The renewal of urban expressways is not simply a matter of renovation and expansion or demolition and reconstruction but of giving them a new vitality through renewal. However, the existing research does not fully consider the harmony, intelligence, resilience, and green characteristics of a road and cannot meet the needs of renewal evaluation. This study focused on whether urban expressways need to be renewed, how urgent the renewal is, and what renewal strategy to adopt. The concept of the urban expressway renewal degree was proposed, and a generic and quantifiable urban expressway renewal indicator system was constructed through three levels: macro, meso, and micro. A hybrid evaluation framework and a renewal degree decision model were constructed to measure the renewal degree of an expressway. This study helps to rationalize the planning of the renewal order of expressways and helps decision-makers formulate scientific renewal plans.

## 2. Literature Review

### 2.1. Evaluation Dimensions of Indicator Systems

In the past decades, both abroad and in China, a large body of research has been published regarding the improvement of road assessment systems [7,11,12]. Anderson et al. [14] focused on the Greenroads system, assessing the permeability of the pavement, recycling of materials and the impact of changes in water runoff around the road while also adding assessments of waste pollution and noise environmental assessment. Hoxha et al. [15] assessed the effect of reducing carbon emissions, energy, and water consumption through the choice of road surface materials and the use of recycled material. Research [16] proposed integrating road construction with the surrounding environment to enhance the

landscape of the road by creating green spaces to play the role of an urban green block; these studies focused on environmental protection during the construction process. Other studies focused on the harmony between the road and the social factors and humanities. To reflect the human-centered thinking, Li et al. [17] fully collected the functional needs of residents for pavements when evaluating road renovation; Istrate, A. [18] attempted to enhance the sense of belonging of the residents and the livability of the street by improving the culture and personality of the street. Ibrahim et al. [19] included factors such as management systems, leadership, and whether traffic development planning is considered in an integrated manner in road management, proposing new requirements from the dimension of facility management. Although the abovementioned studies improved the road assessment system from different dimensions, they still did not meet the renewal needs and most were limited to qualitative evaluation [2,12,20], with the indicators lacking specific quantitative methods.

### 2.2. Evaluation Objectives of Indicator Systems

In terms of evaluation objectives, some studies established a green highway improvement and expansion road evaluation system based on the Greenroads index system for highways, highlighting the goal of environmental protection and considering evaluation criteria such as ecological protection, pollution prevention, resource conservation, energy saving, and emission reduction [11,14,21,22]. El et al. [2] argued that previous studies did not adequately integrate the concept of sustainability into the management of road infrastructure projects and proposed the comprehensive highway sustainability index (CHSI) for assessing the three main categories of sustainable development economics, social criteria, and environmental criteria to improve the level of sustainability in road construction. Similarly, Ibrahim et al. [19] developed a sustainability index for Egyptian highway construction projects that reflects the number of sustainable options implemented during the construction process of highways and even in the maintenance process. Rostamnezhad et al. [23] explored the impact indicators for improving social sustainability in the highway construction process, such as stakeholder participation factors, labor requirements, safety-related factors, and health-related factors and considered the complex interactions between them. The abovementioned studies optimized the indicator system of the three dimensions with the goal of sustainability or focused on one of them for in-depth research. However, in the context of the era of digital infrastructure development, these studies cannot yet be referenced in the process of evaluating the renewal of expressways characterized by intelligence and resilience; therefore, the objectives of renewal can be further investigated.

### 2.3. Renewal Degree Evaluation Model

Many expressways in cities need to be maintained or renovated every year. The question of which expressways have a higher priority for renewal, given a limited budget, needs to be analyzed by a measurable value, and constructing an evaluation model is an important method for sustainable renewal [24]. In the field of community renewal, researchers have evaluated this by constructing a hybrid AHP and technique for order preference by similarity to an ideal solution (TOPSIS) decision model to obtain their renewal priority ranking and improvement strategies [25]. A new spatial decision support system, consisting of a multi-criterion decision-making approach with integrated geographic information systems, was proposed for community-scale urban regeneration projects to measure the regeneration potential of communities [26]. Research [27] proposed a multi-criteria model based on a system dynamics model, which consists of three sub-modules at the city, regional, and community scales and a supporting database that contains both temporal and spatial data to provide decision support for achieving sustainable urban regeneration. Deterministic, probabilistic, and biomechanical models have been constructed to evaluate the priority of road maintenance programs by identifying the three main factors of these programs, namely pavement deterioration, road utility value, and traffic characteristics [28]. By comparing the abovementioned studies, we can claim that at this stage, there have been

more evaluation studies in the field of community renewal but few serving renewal in the field of expressways.

### 2.4. Evaluation Methods

The evaluation methods chosen by scholars are diverse and varied, and there is no consistent evaluation model. Most of the methods currently used are the Delphi method, AHP, the entropy value method, the expert grading method, factor analysis, principal component analysis, the analytic hierarchy process, the analytic network process, the TOPSIS method, the ELECTRE method, data envelopment analysis, the artificial neural network method, the expert system evaluation method, the gray system decision evaluation method, and a combination of methods [29,30]. However, ambiguity and uncertainty in expert opinions still exist in qualitative evaluation methods [31]. Therefore, scholars have combined the fuzzy set theory proposed by Zadeh [32] with the Delphi method to overcome the vagueness and subjectivity of human thinking, judgment, and expression [33]. The fuzzy Delphi method (FDM) is an improvement and enhancement of the classical Delphi method [34]. This approach has been used in a variety of applied fields, including humanities, management, business, physical sciences, and engineering [35]. Based on expert opinions, Huang et al. [36] used the FDM to determine the key input variables that had the deepest impact on blasting vibration. Sumrit et al. [37] used the fuzzy function of triangular fuzzy numbers to screen the important factors to improve the competitiveness of enterprise technological innovation based on the existing research results. Chan et al. [38] combined fuzzy Delphi and analytical network processing techniques to establish a post-disaster resilience index system for redeveloped urban areas in the Tansui River Basin in Taiwan. Wang et al. [39] combined fuzzy theory with online reviews and used probabilistic linguistic term sets and unbalanced trapezoidal cloud models to statistically describe multi-criteria user rating information to help customers choose hotels. The FDM is widely used in multi-criteria decision-making problems. However, the fuzzy language set can be further improved.

The FAHP method is applied to incorporate an expert's bias and subjectivity. This method provides more flexibility to experts while comparing one factor to others [40]. However, the language range of some fuzzy term sets is limited by the value, and the accuracy is low. Yager [41] extended the linguistic term set proposed by Zadeh (1975) by introducing q-rung orthopair fuzzy sets (q-ROFSs), in which the sum of q powers of affiliation and non-affiliation was at most equal to 1. Furthermore, the decision-maker could adjust the value of q based on different risk attitudes, thus making q wider in scope and closer to the real decision-making environment [42]. Studies have shown that q-ROFSs have important value in multi-attribute decision-making processes [42–44].

In summary, it can be seen that the existing evaluation systems and evaluation frameworks are of limited use in the evaluation of urban expressway renewal. The main reasons are as follows. First, evaluation systems for the renewal of urban expressways are lacking, and the existing evaluation systems are mainly focused on green construction or achieving the goal of project sustainability. Research with the evaluation goal of urban expressway renewal that does not have the important characteristics of expressway renewal is lacking. Second, the evaluation dimensions of the existing relevant indicator system are more micro or macro, and the design is not comprehensive enough. Third, the language of experts in the qualitative evaluation process is not sufficiently rigorous and objective, and expert knowledge acquisition methods can be improved further. Fourth, there is no general evaluation framework combining qualitative and quantitative aspects that can be used to measure the renewal degree of urban expressway.

Considering the gaps in the existing research, we attempted to provide a comprehensive evaluation framework to measure the renewal degree of multiple urban expressways. The process was to first identify the key factors affecting urban expressway renewal indicators based on the FDM using more advanced q-ROFSs in research to improve the accuracy of evaluation. Second, a renewal indicator system was established that comprehensively

considered the macroscopic, mesoscopic, and microscopic levels to improve the shortcomings of the expressway renewal evaluation system based on the FAHP method. Then, TOPSIS was used to measure the renewal values of eight expressways in Shanghai, China, classify the renewal intervals, and finally, provide recommendations to help improve the sustainability level of urban expressway renewal.

## 3. Method for Urban Expressway Renewal Degree Evaluation

### 3.1. Framework

In philosophy, "degree" is a quantitative limit to maintain the relative stability of a substance, and when an object reaches a certain marginal quantitative limit, a qualitative change occurs. Based on the characteristics and connotations of sustainable expressway renewal, we propose the concept of the urban expressway renewal degree (UERD), which is a comprehensive evaluation index to measure the necessity (urgency) of urban expressway renewal. Specifically, it is a comprehensive evaluation index that considers the operational status of the road, its compatibility with the social environment, its level of intelligence, its ability to resist risks, and its adaptability to the development trend of the regional environment. The degree of renewal as an expressway evaluation index can help achieve the orderly transformation of the road to be renewed and also identify the key elements of expressway renewal.

Any regeneration is closely linked in time and space. The degree of renewal cannot be discussed simply from the object of transformation but should be placed in the context of the entire urban system; that is, it should fully consider the external effects of the area in which it is located and examine the impact of renewal on the city from a dynamic point of view. Therefore, the development of renewal indicators is based on the time and space environment in which the road is located, and the indicator development strategy is carried out at three levels: macro, meso, and micro. The micro level refers to the need for facility-related renewal, the meso level refers to the need for resilience enhancement, and the macro level refers to the need to adapt to future development. The relationship between them is that the improvement of facility performance leads to the improvement of its ability to respond to natural disasters, thus promoting the overall increase of its ability to serve future social development. A schematic of the urban expressway renewal degree evaluation levels is shown in Figure 1.

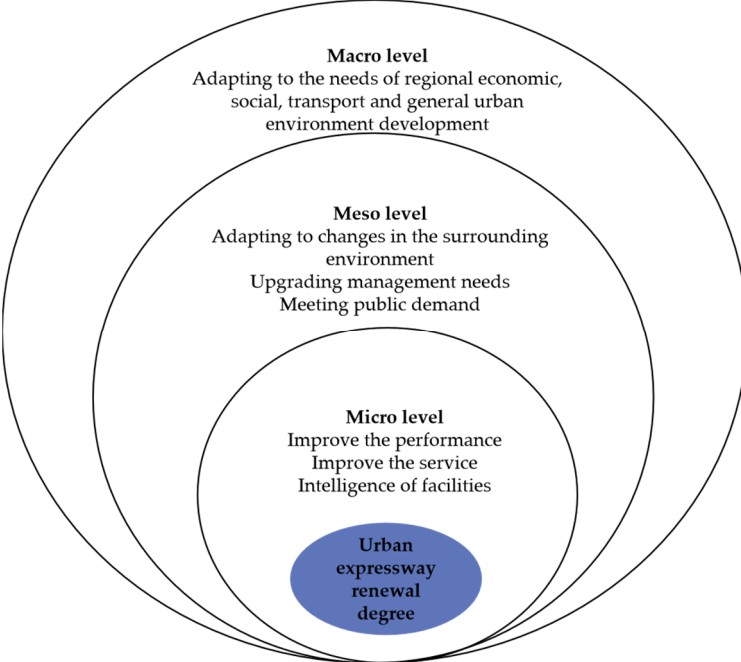

**Figure 1.** Schematic of the renewal degree evaluation hierarchy.

In this research, a method was constructed to fill the gaps in existing research by taking an expressway renewal project in China as an example, as shown in Figure 2. The update degree indicators were screened using the FDM, the indicators were weighted using the FAHP method, and the update degree values were measured using the TOPSIS-based method.

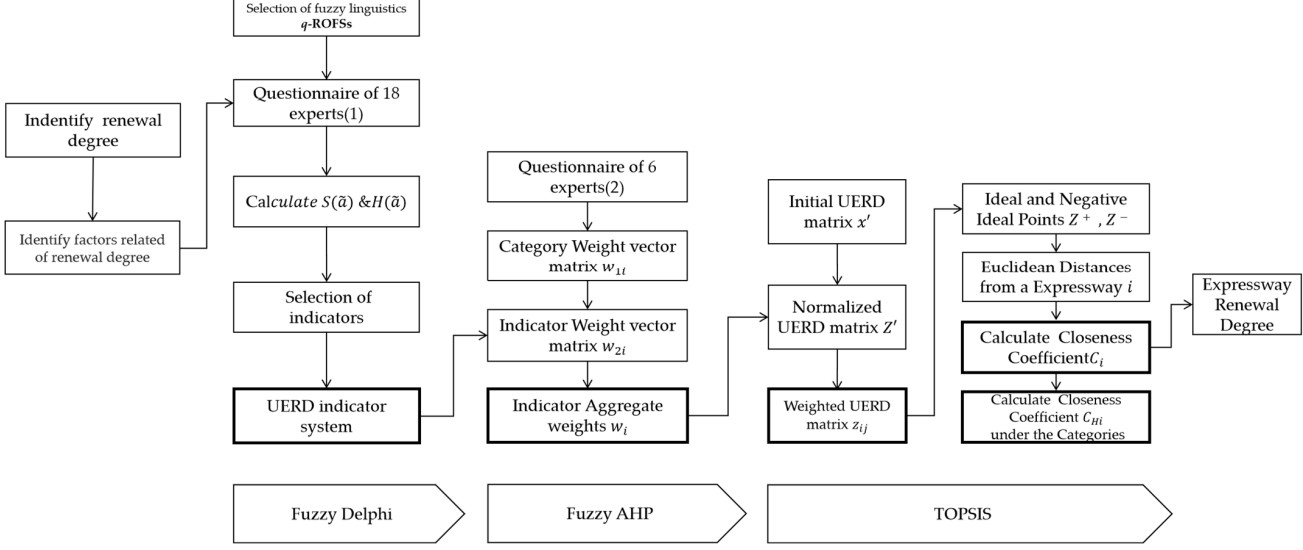

**Figure 2.** Flowchart of the research methodology.

### 3.2. Categories and Sources of Primary Indicators

Based on the above framework of macro, meso, and micro levels, we initially set up eight categories of renewal evaluation indicators, and based on literature research, case studies, and an expert brainstorming method, we obtained 32 corresponding influencing factors. These are shown in Table 1.

**Table 1.** Initial factors of UERD.

| Criteria | Indicators | References |
|---|---|---|
| 1. Facility | Hardware facilities<br>Facility maintenance<br>Driving safety<br>Classification of roads<br>Length of road | [28,45] |
| 2. Service functionality | Road saturation<br>Traffic accessibility<br>Level of intelligent control<br>Public satisfaction<br>Road importance | [17,18,21,46] |
| 3. Physical resilience | Flooding resilience<br>Fog resilience<br>Storm resilience<br>Earthquake resilience<br>Landslide resilience<br>Road surface icing resilience<br>Fire resilience | [47] |
| 4. Institutional resilience | Emergency response to major disasters<br>Energy conservation management system<br>Accessibility of information | [45,48] |

**Table 1.** *Cont.*

| Criteria | Indicators | References |
|---|---|---|
| 5. Economic demand | Policy direction<br>Gross domestic product<br>Regional infrastructure investment | [12,49,50] |
| 6. Transportation demand | Cross-regional traffic flow<br>Industrial layout requirements<br>Characteristic development needs | [20,51] |
| 7. Social demand | Population growth<br>Demographic structure<br>Employment position | [12] |
| 8. Environmental demand | Environmental pollution assessment<br>Carbon emission assessment<br>Harmony with the urban environment | [15,20,21,52] |

*3.3. Selection of Performance Indicators Based on FDM*

3.3.1. q-Rung Orthopair Fuzzy Sets

This study used the set of *q*-ROFS language proposed by Yager for fuzzy evaluation, with the relevant definitions as follows:

**Definition 1.** [41]. Let $X = \{X_1, X_2, \ldots, X_n\}$ be a universe of discourse; then, a *q*-ROFS $\widetilde{A}$ defined on $X$ is given by

$$\widetilde{A} = \{\langle\, x, \mu_{\widetilde{A}}(x), v_{\widetilde{A}}(x)\rangle | x \in X \} \tag{1}$$

where $\mu_{\widetilde{A}} : X \to [0,1]$ indicates the degree of membership, and $v_{\widetilde{A}} : X \to [0,1]$ indicates the degree of non-membership of $X$ to $\widetilde{A}$, satisfying the condition $0 \leq \mu_{\widetilde{A}}(x) \leq 1$, $0 \leq v_{\widetilde{A}}(x) \leq 1$, $0 \leq \left(\mu_{\widetilde{A}}(x)\right)^q + \left(v_{\widetilde{A}}(x)\right)^q \leq 1, q \geq 1$. The degree of indeterminacy of $X$ to $\widetilde{A}$ is denoted as $\pi_{\widetilde{A}}(x) = \left(1 - \left(\mu_{\widetilde{A}}(x)\right)^q - \left(v_{\widetilde{A}}(x)\right)^q\right)^{1/q}$. For simplicity, the pair $\widetilde{a} = (\mu, v)$ is called a *q*-rung orthopair fuzzy number (*q*-ROFN).

**Definition 2.** [42]. Let $\widetilde{a} = (\mu, v)$ be a q-ROFN; then, the score function $S(\widetilde{a})$ and the accuracy function of $H(\widetilde{a})$ are defined as follows:

$$S(\widetilde{a}) = \mu^q - v^q \tag{2}$$

$$H(\widetilde{a}) = \mu^q + v^q \tag{3}$$

It is clear that $S(\widetilde{a}) \in [-1, 1]$ and $H(\widetilde{a}) \in [0, 1]$.

**Definition 3.** [42]. Let $\widetilde{a} = (\mu, v)$, $\widetilde{a}_1 = (\mu_1, v_1)$, and $\widetilde{a}_2 = (\mu_2, v_2)$ be any three *q*-ROFNs; then, the basic operations of *q*-ROFNs are as follows:

$$\widetilde{a}_1 \oplus \widetilde{a}_2 = \left(\left(\mu_1^q + \mu_2^q - \mu_1^q\mu_2^q\right)^{1/q}, v_1 v_2\right) \tag{4}$$

$$\widetilde{a}_1 \otimes \widetilde{a}_2 = \left(\mu_1\mu_2, \left(v_1^q + v_2^q - v_1^q v_2^q\right)^{1/q}\right) \tag{5}$$

$$\lambda\widetilde{a} = \left(\left(1 - (1 - \mu^q)^\lambda\right)^{1/q}, v^\lambda\right), \lambda > 0 \tag{6}$$

$$\widetilde{a}_\lambda = \left(\mu^\lambda, \left(1 - (1 - v^q)^\lambda\right)^{1/q}\right), \lambda > 0 \tag{7}$$

**Definition 4.** [42]. Let $\widetilde{a}_1 = (\mu_1, v_1), \widetilde{a}_2 = (\mu_2, v_2)$ be any two $q$-ROFNs; the comparison rules of $q$-ROFNs are as follows:

$$(1) \text{ If } S(\widetilde{a}_1) > S(\widetilde{a}_2), \text{ then } \widetilde{a}_1 > \widetilde{a}_2 \tag{8}$$

$$(2) \text{ If } S(\widetilde{a}_1) = S(\widetilde{a}_2), \text{ then}$$

$$(a) \text{ If } H(\widetilde{a}_1) > H(\widetilde{a}_2), \text{ then } \widetilde{a}_1 > \widetilde{a}_2 \tag{9}$$

$$(b) \text{ If } H(\widetilde{a}_1) = H(\widetilde{a}_2), \text{ then } \widetilde{a}_1 = \widetilde{a}_2 \tag{10}$$

**Definition 5.** [42]. Let $\widetilde{a}_i = (\mu_i, v_i)(i = 1, 2, \ldots, n)$ be a collection of $q$-ROFNs and $w = (w_1, w_2, \ldots, w_n)^T$ be the weight vector of $\widetilde{a}_i(i = 1, 2, \ldots, n)$ with $w_i \in [0,1]$ and $\sum_{i=1}^{n} w_i = 1$. Then, the $q$-rung orthopair fuzzy weighted averaging ($q$-ROFWA) operator is defined as follows:

$$q - ROFWA(\widetilde{a}_1, \widetilde{a}_2, \ldots, \widetilde{a}_n) = \overset{n}{\underset{i=1}{\oplus}} w_i\widetilde{a}_i = \left( \left(1 - \prod_{i=1}^{n}\left(1 - \mu_i^q\right)^{w_i}\right)^{\frac{1}{q}}, \prod_{i=1}^{n} v_i^{w_i}\right), \tag{11}$$

3.3.2. Fuzzy Delphi Method

The standard computation procedure for executing the FDM is as follows:

(1) Experts are requested to rate each factor of UERD based on their importance using a linguistic scale, as shown in Table 2; this scale helps capture the opinions of experts by $q$-ROFN and is represented as follows: $\widetilde{a} = (\mu, v)$.

(2) The score function and the accuracy function calculations use Equations (2) and (3).

(3) The relevant factors are traded off based on the magnitude of the score function $S(\widetilde{a}_i)$, and when two factors have equal score function values, they are compared using Equations (9) and (10). A value is also set as a criterion; that is, when the value of $S(\widetilde{a}_i)$ is below $-0.1$, the indicator is rejected.

**Table 2.** Linguistic scales for the FDM.

| Linguistic Terms | q-ROFNs ($\mu$,$v$) |
|---|---|
| Weakly Important (WI) | (0.30, 0.85) |
| Equally Important (EI) | (0.50, 0.50) |
| Moderately Important (MI) | (0.75, 0.40) |
| Important (I) | (0.80, 0.25) |
| Very Important (VI) | (0.95, 0.10) |

If $S(\widetilde{a}_i) \geq 0.1$, then factor $i$ is selected.

If $S(\widetilde{a}_i) < 0.1$, then factor $i$ is rejected.

After the literature research, we collected 32 factors that affect the sustainability of expressways and included these 32 factors in a questionnaire survey to collect expert ratings. During the research process, 18 experts with more than 5 years of working experience in this field were invited to conduct the analysis. The experts' knowledge backgrounds were as follows: (1) professors or academics working in the field of urban transport infrastructure and (2) experts working in the field of planning, construction, or operation. Their professional titles were as follows: six senior engineers, four professors, three associate professors, one chief engineer of a design institute, one chief planner of a design institute, one design engineer and two engineers.

In the questionnaire, the experts were asked to evaluate three aspects of the road renewal indicators: (1) to judge and rate the necessity and relevance of each factor in the initial evaluation system to check whether it is consistent with the evaluation objectives; (2) to seek expert advice on this indicator system as a corresponding supplement; and (3) to evaluate the rationality of the quantification method for each factor and provide

suggestions for improvement. Through careful examination of the returned questionnaires and comprehensive calculation, we finally rejected four factors and screened out 28 factors (Table 3). Moreover, the quantification methods of eight indicators were improved to make them more feasible.

**Table 3.** Initial indicators' evaluation results.

| Clusters | Indicators | Fuzzy Weights | $S(\tilde{a})$ | Decision |
|---|---|---|---|---|
| Facility renewal requirements (H1) | Hardware facilities (C1) | (0.825, 0.000) | 0.561 | Accepted |
| | Facilities maintenance (C2) | (0.825, 0.000) | 0.561 | Accepted |
| | Driving safety (C3) | (0.789, 0.000) | 0.490 | Accepted |
| | Classification of roads | (0.077, 0.000) | 0.000 | Rejected |
| | Length of road | (0.075, 0.000) | 0.000 | Rejected |
| Service functionality renewal requirements (H2) | Road saturation (C4) | (0.789, 0.000) | 0.490 | Accepted |
| | Traffic accessibility (C5) | (0.819, 0.000) | 0.549 | Accepted |
| | Level of intelligent control (C6) | (0.819, 0.000) | 0.550 | Accepted |
| | Public satisfaction (C7) | (0.630, 0.000) | 0.251 | Accepted |
| | Road importance (C8) | (0.649, 0.000) | 0.273 | Accepted |
| Physical resilience (H3) | Flooding resilience | (0.075, 0.000) | 0.000 | Rejected |
| | Fog resilience | (0.182, 0.000) | 0.006 | Rejected |
| | Storm resilience (C9) | (0.649, 0.000) | 0.273 | Accepted |
| | Earthquake resilience (C10) | (0.817, 0.000) | 0.546 | Accepted |
| | Landslide resilience (C11) | (0.825, 0.000) | 0.561 | Accepted |
| | Road surface icing resilience (C12) | (0.825, 0.000) | 0.561 | Accepted |
| | Fire resilience (C13) | (0.825, 0.000) | 0.561 | Accepted |
| Institutional resilience (H4) | Efficiency of emergency response to major disasters (C14) | (0.776, 0.000) | 0.467 | Accepted |
| | Energy conservation management system (C15) | (0.803, 0.000) | 0.518 | Accepted |
| | Accessibility of information (C16) | (0.803, 0.000) | 0.518 | Accepted |
| Economic development needs (H5) | Policy orientation (C17) | (0.814, 0.000) | 0.540 | Accepted |
| | Gross domestic product(C18) | (0.803, 0.000) | 0.518 | Accepted |
| | Regional infrastructure investment (C19) | (0.813, 0.000) | 0.538 | Accepted |
| Demand for transportation development (H6) | Cross-regional traffic flow (C20) | (0.825, 0.000) | 0.561 | Accepted |
| | Industrial layout requirements (C21) | (0.721, 0.000) | 0.375 | Accepted |
| | Characteristic development needs (C22) | (0.825, 0.000) | 0.561 | Accepted |
| Social development needs (H7) | Population growth (C23) | (0.825, 0.000) | 0.561 | Accepted |
| | Demographic structure (C24) | (0.751, 0.000) | 0.424 | Accepted |
| | Employment position (C25) | (0.825, 0.000) | 0.561 | Accepted |
| Environmental development needs (H8) | Environmental pollution assessment (C26) | (0.814, 0.000) | 0.540 | Accepted |
| | Carbon emission assessment (C27) | (0.819, 0.000) | 0.549 | Accepted |
| | Harmony with the urban environment (C28) | (0.353, 0.000) | 0.044 | Accepted |

### 3.3.3. Urban Expressway Renewal Degree Indicator System

Based on the concept of the regeneration index system described in Section 3.1, combined with the evaluation results in Section 3.3.2, the UERD indicators were divided into three major categories: the need for facility renewal, the need for resilience enhancement, and the need to adapt to future development. There were eight primary criteria and 28 indicators.

(1) Facility renewal requirements

At the micro level, the performance of the expressway facility was our main concern. Facility renewal requirements were evaluated in terms of structural renewal requirements (H1) and service renewal requirements (H2). The structural renewal requirements (H1) were evaluated in terms of the engineering condition of the facility, the operational maintenance condition, and the safety of the traffic. The engineering condition could be evaluated with the morphological quality index (MQI) and the road quality index (RQI). In the course of operation and maintenance, the road has been regularly maintained by the relevant management department every year, its performance was evaluated by the relevant management department, and the above data could be obtained from the performance evaluation report. The safety of traffic can reflect the overall condition of the facility structure to a certain extent, with higher accident rates correlating with the aging of the

facility structure. This can be described by the number of traffic accidents per kilometer, per year on the road section, with the specific data coming from the road management platform data.

Service function reflects the comfort level of road facilities for vehicle operation and resident travel services. According to the technical standards for road engineering (JTG B01-2014), road saturation can reflect the traffic load of the road during peak hours, calculated by the ratio of the actual traffic volume to the design traffic volume. The smoothness of traffic reflects the congestion time of traffic and is evaluated with the road traffic index. The level of intelligent control reflects to a certain extent the degree of intelligence of the facility, which is in line with a newer direction of intelligent infrastructure construction, as quantified by Table 4, which shows the above quantitative data source road management platform. The importance of the road reflects the importance of the road in the region, the traffic zoning, and functional zoning to reflect the quantitative data from the government public data. Public satisfaction is also a manifestation of the level of service function, which could be obtained from the performance evaluation report or through questionnaire surveys.

**Table 4.** Facility renewal requirement indicators.

| Field | Criteria | Indicators | Description/Measurement Method | Quantitative Method |
|---|---|---|---|---|
| Facility renewal requirements | Expressway condition renewal requirements (H1) | Hardware facilities (C1) | Status of road engineering | Morphological quality index and quality index of the road |
| | | Facilities maintenance (C2) | Completion of the annual maintenance plan for roads, carried out by the operations department | Conservation performance evaluation report score |
| | | Driving safety (C3) | Number of traffic accidents per kilometer on the road (years) | Number of accidents (number/km-yr) |
| | Service level (H2) | Road saturation (C4) | Ratio of actual road traffic to design traffic during peak hours | V/C = Actual traffic volume/design traffic volume |
| | | Traffic accessibility (C5) | Road traffic index | Peak congestion hours/smooth passage times |
| | | Level of intelligent control (C6) | Refers to the information collection, processing and dissemination efficiency, emergency dispatching capability, and linked disposal capability of traffic accidents | Number of video surveillance cameras and sensors per kilometer (units/km) |
| | | Public satisfaction (C7) | Passing vehicle satisfaction and public hotline disposal | Questionnaire score |
| | | Road importance (C8) | Reflected by the importance of the area in which it is located, the traffic zoning in which it is located, and the functional zoning | Weighted calculation of the importance of nodes at both ends of the road |

(2) The need for resilience enhancement

At the meso level, the impact of changes in the natural environment in which it is located on road facilities was one of the most important factors [12]. Based on the definition and classification of resilience by Ostadtaghizadeh (2015) [47], we recognize that natural resilience (H5) refers to the ability of expressways to resist natural disasters. Natural disasters facing expressways include heavy rainfall, earthquakes, rainy season landslides, road surface icing, and fire. The degree of damage to expressways and ancillary facilities caused by these disasters each year determines the length of their maintenance time, and the target is to improve the resilience of the expressway to disasters and its ability to recover from them. This indicator could be quantified by the number and extent of disasters affecting expressway traffic each year. The quantitative data were derived from management statistics.

Expressways are managed by multiple departments, and the need to update institutional resilience (H4) reflects the level of management of expressways from the perspective of facility management. The institutional resilience of expressways includes the ability to deal with major disasters, the energy-saving management system, and the smoothness of information channels. The ability to respond to and deal with major disasters is reflected in the soundness of the emergency response plan, whether all relevant departments have a coordinated emergency management mechanism, whether there is a multi-departmental

joint information dissemination mechanism, and whether there is an emergency rescue plan, all of which affect the timeliness of dealing with major accidents when they occur [48]. Based on the available statistical data, the average handling time of major incidents per year was used to quantify the data from the management platform. The energy-saving management system serves to achieve energy saving and emission reduction on expressways and has been quantified with a rating by experts. The smoothness of information channels fully reflects the degree of public participation in the construction of expressways and the humanistic temperature of the city, and it was calculated according to the rate of completion of the disposal of opinions and suggestions by the management of this road, with quantitative data coming from the performance evaluation report. See Table 5 for details.

**Table 5.** Demand for resilience enhancement indicators.

| Field | Criteria | Indicators | Description/Measurement Method | Quantitative Method |
|---|---|---|---|---|
| Demand for resilience enhancement | Natural resilience (H3) | Storm resilience (C9) | Number of times and the extent to which traffic is affected by extreme weather or sudden natural disasters (e.g., heavy rainfall, earthquakes, rainy season landslides, and icy roads) on road sections (main lines and gateways) (per year) | Shutdown time per km per year (h/km) |
| | | Earthquake resilience (C10) | | |
| | | Landslide resilience (C11) | | |
| | | Road surface icing resilience (C12) | | |
| | | Fire resilience (C13) | Refers to the extent of damage to expressways and ancillary facilities resulting from a fire | Fire-induced shutdown time (h/year) |
| | Institutional resilience (H4) | Efficiency of emergency response to major disasters (C14) | The efficiency of emergency response and incident management after an incident has occurred | Average processing time for major incidents (h/year) |
| | | Energy conservation management system (C15) | -Whether an organizational structure and related systems are set up for energy conservation and emission reduction -Whether a statistical monitoring system is in place -Whether an assessment system is in place -Whether public awareness and training campaigns are conducted | Grade rating (1–5) |
| | | Accessibility of information (C16) | Refers to timely feedback and disposition of public suggestions | Completion rate of comment and suggestion disposal (%) |

(3) Adapting to future development needs

At the macro level, expressways were placed in the context of the general environment of the region in which they are located and were evaluated to see if they meet the needs of future sustainable development. According to the sustainable development theory, the evaluation was mainly performed in three dimensions: economic development demand, social development demand, and environmental development demand. To reflect the development characteristics of the expressway, this study added the traffic development demand [49]. Economic development demand (H5) refers to the rapid development of the region's economy as a result of the government's new development plan or the promulgation of new policies conducive to the region's economic development; this indicator was calculated quantitatively through policy orientation, the regional gross domestic product, and the regional investment amount.

The demand for traffic development (H6) reflects the degree of interregional and intercity traffic movement and circulation and was quantified by the traffic flow statistics at the gates of the expressway management platform. It also includes the traffic demand caused by the influence of the regional industrial layout structure and the establishment of special sections, which was quantified through the classification of classes.

Social development demand (H7) refers to the growth in traffic demand caused by the growth in regional population, demographics, and job creation. Quantifiable data were obtained from regional statistical yearbooks or government statistical reports.

Environmental development demand (H8) includes environmental pollution evaluation and carbon emission evaluation. The quantifiable data came from the regional ecological and environmental bureau statistical reports. Harmonization with the urban environment was achieved via a qualitative evaluation, with data from questionnaires. The details of the above descriptions are presented in Table 6.

**Table 6.** Adaptation to future discovery needs indicators.

| Field | Criteria | Indicators | Transfer Function/Questionnaire | Quantitative Method |
|---|---|---|---|---|
| Adapting to future development needs | Economic development needs (H5) | Policy direction (C17) | Government documents and planning documents that are conducive to the development of the district | Number of relevant government public documents (pcs) |
| | | Gross domestic product (C18) | Gross regional product growth rate | Average regional GDP growth rate over the last 3 years (%) |
| | | Regional infrastructure investment (C19) | Area investment per square kilometer | Investment/square kilometer (million yuan/square kilometer) |
| | Transport development needs (H6) | Cross-regional traffic flow (C20) | Transit traffic flows, that is, the proportion of traffic flows passing through a region that do not have an origin or destination in that region | Transit traffic/total traffic (%) |
| | | Industrial layout requirements (C21) | The availability of industrial restructuring | Grade rating (0,1) |
| | | Characteristic development needs (C22) | The presence or absence of significant development blocks along the route or in the region -Newly assigned district functions -Characteristic development plans | Grading scale (1–5) |
| | Social development needs (H7) | Population growth (C23) | Population growth in the last 3 years in the region | Three-year growth rate of regional population (%) |
| | | Demographic structure (C24) | Percentage of population aged 15–59 years | Percentage of population aged 15–59 years (%) |
| | | Employment position (C25) | Employment growth rate in the last 3 years in your region | Three-year average growth rate of jobs (%) |
| | Environmental development needs (H8) | Environmental pollution assessment (C26) | Air noise and water pollution | Environmental assessment score |
| | | Carbon emission assessment (C27) | ETC coverage Energy-efficient lighting and other green, low-carbon technologies Non-stop overload pre-screening system Intelligent control of tunnel lighting and ventilation -Renewable energy utilization Energy consumption monitoring and management platform construction | Grade rating (1–5) |
| | | Harmony with the urban environment (C28) | Uniformity of appearance with the city's historical and cultural landscape Uniformity of appearance with the modern development of the city | Grade rating (1–5) |

### 3.4. Index Weighting Based on the FAHP Method

Studies have shown that FAHP exhibits better robustness and sensitivity to multi-attribute decision problems [53]. The specific steps of the FAHP implementation were as follows.

Step 1. Assign weight to the experts. The number of experts is $l$. Each expert was assigned a weight based on their knowledge base, work experience, and confidence in completing the questionnaire $\lambda_k$ and $\lambda_k > 0$, $\sum_{k=1}^{l} \lambda_k = 1$.

Step 2. Evaluate the importance of the first-level indicators. The experts used a linguistic scale (Table 7) to evaluate the first-level indicators. Very high is the most important indicator of the target level, and very low is the opposite. This resulted in a matrix for the evaluation of indicators $\left( \widetilde{w}_k = \left( \mu_{wi}^k, v_{wi}^k \right) \right)$.

**Table 7.** Linguistic scales for the FAHP.

| Linguistic Terms | q-ROFNs ($\mu$,$\nu$) |
|---|---|
| Very Low (VL) | (0.15, 0.90) |
| Low (L) | (0.30, 0.85) |
| Medium Low (ML) | (0.45, 0.65) |
| Medium (M) | (0.50, 0.50) |
| Medium High (MH) | (0.75, 0.40) |
| High (H) | (0.80, 0.25) |
| Very High (VH) | (0.95, 0.10) |

Step 3. After obtaining the individual indicator weights, calculate the combined q-ROFs fuzzy weights of the indicators as follows:

$$\widetilde{w}_k = (\mu_{wi}, \nu_{wi}) = q - ROFWA\left(\widetilde{w}_i^1, \widetilde{w}_i^2, \ldots, \widetilde{w}_i^l\right)$$
$$= \left(\left(1 - \prod_{k=1}^{l}\left(1 - \left(\mu_{wi}^k\right)^q\right)^{\lambda_k}\right)^{1/q}, \prod_{k=1}^{l}\left(\nu_{w_i}^k\right)^{\lambda_k}\right) \tag{12}$$

Step 4. Calculate the final weights of the indicators.

The final weights of the primary indicators $W = (w_1, w_2, I, w_m)$ were calculated as follows:

$$w_{1i} = \frac{S(\widetilde{w}_i)}{\sum_{i=1}^{m} S(\widetilde{w}_i)}, i = 1, 2, \ldots, m \tag{13}$$

Similarly, the weights of the secondary indicators could be obtained as $\widetilde{w}_{2i} = (w_{21}, w_{22}, \ldots, w_{2n})$.

Step 5. Defuzzify the indicators using Formula (2) to obtain the score function values for the primary and secondary indicator weights, $w_{1i}, w_{2i}$.

Step 6. Aggregate the primary indicators with the secondary indicators to obtain the final weights of the secondary indicators' aggregation operator.

$$w_i = w_{1i} \times w_{2i} \tag{14}$$

In this stage, the FAHP method was applied to obtain the primary and secondary indicator weights and aggregation weights. Six experts with more than 10 years of working experience in this field in the industry were invited to perform the evaluation, two of these experts are university professors and four are senior engineers. Calculate the weight vector according to the evaluation score of expert's confidence, and the weight vector of the experts was $\lambda_k = 0.18, 0.15, 0.14, 0.21, 0.13.0.19$, and the evaluation process used a fuzzy linguistic scale (Table 7) to compare and score indicators two by two, where the original evaluation matrix was obtained ($M$). Using this matrix, we calculated the aggregated weights for each level of indicators ($M_1$), ($M_2$) using Equation (12). The final weights of the indicators were then calculated using Equation (13). Table 8 presents the weight values of the primary ($w_{1i}$) and secondary ($w_{2i}$) indicators obtained using the FAHP method and the order in which they were ranked. The aggregated weights of the secondary indicators and their ranking were then obtained using Equation (14), with the final aggregated weight ($w_i$) being calculated by multiplying the primary weights with their respective secondary indicator weights.

**Table 8.** Weighting and ranking of indicators at each level.

| Criteria | Weights | Rank | Indicators | Weights | Rank | Aggregate Weights | Final Rank |
|---|---|---|---|---|---|---|---|
| H1 | 0.120 | 4 | C1 | 0.329 | 10 | 0.039 | 11 |
| | | | C2 | 0.256 | 16 | 0.031 | 18 |
| | | | C3 | 0.415 | 3 | 0.050 | 5 |
| H2 | 0.148 | 2 | C4 | 0.211 | 20 | 0.031 | 16 |
| | | | C5 | 0.210 | 21 | 0.031 | 17 |
| | | | C6 | 0.160 | 28 | 0.024 | 22 |
| | | | C7 | 0.196 | 25 | 0.030 | 20 |
| | | | C8 | 0.222 | 17 | 0.033 | 14 |
| H3 | 0.099 | 8 | C9 | 0.189 | 26 | 0.019 | 27 |
| | | | C10 | 0.214 | 19 | 0.021 | 24 |
| | | | C11 | 0.177 | 27 | 0.018 | 28 |
| | | | C12 | 0.210 | 22 | 0.021 | 25 |
| | | | C13 | 0.210 | 23 | 0.021 | 26 |
| H4 | 0.126 | 3 | C14 | 0.469 | 1 | 0.059 | 2 |
| | | | C15 | 0.198 | 24 | 0.025 | 21 |
| | | | C16 | 0.333 | 9 | 0.042 | 9 |
| H5 | 0.114 | 5 | C17 | 0.349 | 7 | 0.040 | 10 |
| | | | C18 | 0.391 | 5 | 0.045 | 7 |
| | | | C19 | 0.260 | 15 | 0.030 | 19 |
| H6 | 0.180 | 1 | C20 | 0.382 | 6 | 0.069 | 1 |
| | | | C21 | 0.314 | 11 | 0.057 | 3 |
| | | | C22 | 0.304 | 13 | 0.055 | 4 |
| H7 | 0.106 | 7 | C23 | 0.446 | 2 | 0.047 | 6 |
| | | | C24 | 0.220 | 18 | 0.023 | 23 |
| | | | C25 | 0.334 | 8 | 0.035 | 12 |
| H8 | 0.108 | 6 | C26 | 0.392 | 4 | 0.042 | 8 |
| | | | C27 | 0.314 | 12 | 0.034 | 13 |
| | | | C28 | 0.295 | 14 | 0.032 | 15 |

*3.5. TOPSIS Method for the Order of Preference by Similarity to an Ideal Solution*

TOPSIS is one of the most commonly used methods for measuring the distance between evaluation solutions and ideal solutions [54], and it has the advantage of solving problems such as the prioritization and ranking of large-scale evaluation objects in a quantitative way [55]. It measures the distance to the best solution (ideal) but also to the anti-ideal. Therefore, it is commonly used in the field of engineering management for solution comparison and evaluation [56,57]. This study used this method for the evaluation, ranking, and comparison of expressways through the following steps.

Step 1. Standardizing indicators

Owing to the different units and sizes of the acquired data, all of the indicators had to be converted into a uniform measurement scale. The evaluation indicators were divided into very large indicators and very small indicators. First, the very small indicators were normalized using the following formula:

$$x' = max - x \tag{15}$$

Next, all of the indicators were normalized by the linear scale transformation (Max–Min) method:

$$Z_{ij} = \frac{x'_{ij}}{max(x') - min(x')} \tag{16}$$

$$Z'_{ij} = \frac{x_{ij}}{\sum_{i=1}^{j} x_{ij}} \tag{17}$$

Step 2. Weighting of normalized indicators

The weighted value of the normalized indicator ($Z\prime_{ij}$) was calculated as follows:

$$z_{ij} = w_i Z'_{ij} \tag{18}$$

where $w_i$ is the weight of the indicator, and $Z_{ij}$ is the normalized value of the indicator.

Step 3. Ideal and negative ideal points

The positive ideal point ($Z^+$) is the combination of the best performance values among all of the indicators, while the negative ideal point ($Z^-$) is the combination of the worst performance values. They were determined by the following equations.

$$z_j^+ = \left( max(z_{11}, z_{21}, \ldots, z_{n1}), max(z_{12}, z_{22}, \ldots, z_{n2}), \ldots, max\left(z_{1j}, z_{2j}, \ldots, z_{nm}\right)\right), \tag{19}$$

$$z_j^- = \left( min(z_{11}, z_{21}, \ldots, z_{n1}), min(z_{12}, z_{22}, \ldots, z_{n2}), \ldots, min\left(z_{1j}, z_{2j}, \ldots, z_{nm}\right)\right) \tag{20}$$

Step 4. Euclidean distance from sample $j$ to the ideal point

The Euclidean distance from the road $j$ to be updated to the positive ideal point ($z^+$) and the negative ideal point ($z^-$) was calculated as follows:

$$D_j^+ = \sqrt{\sum_{j=1}^n \left(z_j^+ - z_{ij}\right)^2}, j = 1, 2, \ldots, m, \tag{21}$$

$$D_j^- = \sqrt{\sum_{j=1}^n \left(z_{ij} - z_j^-\right)^2}, j = 1, 2, \ldots, m \tag{22}$$

Step 5. Calculating the closeness coefficient

The value of the closeness coefficient $C_j$ is used to indicate the relative proximity of the evaluation item $i$ to the negative ideal point. The greater the proximity, the further the item is from the negative ideal point and the closer it is to the positive ideal point, the higher its evaluation value.

$$C_j = \frac{D_j^-}{D_j^+ + D_j^-} \tag{23}$$

The expressway renewal degree value is denoted by $C_j$. Similarly, the abovementioned method could be applied to calculate the $C_{Hi}$ values for each type of indicator of the project separately, which represent, for example, the structural renewal demand ($C_{H1}$), the service function renewal demand ($C_{H2}$), and the natural toughness enhancement demand ($C_{H3}$).

## 4. Urban Expressway Renewal Degree Evaluation Application

### 4.1. Data Sources

The Shanghai administration in China published a document related to the promotion of urban infrastructure renewal in 2021, which continuously promotes the renewal of urban expressways. In this study, eight expressways in the city with a long construction life were selected for renewal evaluation, namely the S20 West Section of the Outer Ring Road (Ex1), the G50 Shanghai–Chongqing Expressway (Jiamin Elevated Road—G318) (Ex2), the Elevated Inner Ring Road (Siping Road–Zhengben Road) (Ex3), the Jialiu section of the G15 Expressway (Ex4), the Jiajin section of the G15 Expressway (Ex5), the Yanggao Road (Zhouhai Road–Jinhai Road) (Ex6), the Husong Road (Jiamin Elevated Road—S32 Highway) (Ex7), and the Jiyang Road (Lupu Bridge–Minhang District boundary) (Ex8). See Appendix A for related information.

The data on the expressways were collected based on the quantification method of indicators described in Section 3.3.3. The specific data were obtained from statistical yearbooks, government portals, traffic white papers, data centers of expressway operation and management departments, and expressway performance evaluation reports. The raw

data for the eight expressways from 2019 to 2021 were finally collected. The specific sources of data are shown in Table 9.

**Table 9.** Quantitative data sources for each indicator.

| Indicators | Original Data | Source Channels |
|---|---|---|
| C1, C2, C7 | Expressway operations and maintenance evaluation reports | Expressway-owned O&M company |
| C3, C4, C5, C9, C10, C11, C12, C13, C14, C20 | Traffic network monitoring data | Shanghai Tunnel Co. |
| C6 | Traffic network control center data | Shanghai Tunnel Co. |
| C8, C18, C19, C21, C22, C23, C24, C25 | Government planning reports, statistical yearbook data, and map data | Shanghai Municipal Government Development Planning Report, Shanghai Municipal Bureau of Statistics Yearbook, and Google Maps |
| C15, C26, C27 | Sectoral data | Shanghai Ecological Environment Bureau |
| C16 | Traffic hotline data | Shanghai Traffic Management Department |
| C17 | Government sector data | Official government website |
| C28 | Questionnaire statistics | Questionnaires |

### 4.2. Calculate the Expressway Renewal Degree Based on TOPSIS

This section describes the use of the TOPSIS model to calculate the renewal degree values for expressways. First, the type of indicator was specified and labeled (P and N) according to its characteristics as a positive indicator (benefit-based indicator) or as a negative indicator (cost-based indicator). Next, the data obtained above were calculated according to the way the indicators were quantified, and each indicator for each item was filled in the table to obtain the original matrix ($M_1$). In the next step, the raw matrix data were labeled and normalized to obtain the normalized matrix ($M_s$). Then, the indicator weights ($w$) derived in Section 3.4 were aggregated with the standardized matrix to obtain the assignment matrix ($M_w$).

In the next step, the PIS ($z^+$) and NIS ($z^-$) were calculated for each road. The distances ($d^+, d^-$) between PIS ($z^+$) and NIS ($z^-$) for each road were calculated separately using Equations (21) and (22), as shown in Appendix B. From these distances, the closeness coefficient ($C_i$) was calculated for each road using Equation (23). Finally, the cities were ranked according to the decreasing order of proximity coefficients, as shown in Table 10.

**Table 10.** Closeness coefficient of the expressways and ranking.

| Expressway | $d_i^+$ | $d_i^-$ | $C_i$ | Rank |
|---|---|---|---|---|
| Ex4 | 0.020 | 0.034 | 0.628 | 1 |
| Ex2 | 0.027 | 0.025 | 0.480 | 2 |
| Ex7 | 0.030 | 0.021 | 0.413 | 3 |
| Ex1 | 0.031 | 0.020 | 0.388 | 4 |
| Ex6 | 0.033 | 0.019 | 0.364 | 5 |
| Ex5 | 0.033 | 0.017 | 0.341 | 6 |
| Ex3 | 0.036 | 0.015 | 0.300 | 7 |
| Ex8 | 0.038 | 0.011 | 0.198 | 8 |

Next, the assessment status was determined. According to the warning levels corresponding to different colors in the disaster forecast and the standard division of the correlation degree [58], the degree of urgency of expressway renewal was divided into four intervals (i.e., the blue renewal interval, yellow renewal interval, orange renewal interval, and red renewal interval), and their score intervals are shown in Table 11. According to the grade division of the interval value where the target value was located, the renewal urgency of the expressway and the renewal status result were obtained (Figure 3).

**Table 11.** Update interval division table.

| UERD Interval | Interval Value | Suggestions for Renewal |
|---|---|---|
| Blue interval | [0.00, 0.20] | Repair and maintenance: further maintain the existing advantageous conditions and closely monitor changes in the indicators of renewal needs to enable them to continue healthy operation |
| Yellow interval | [0.20, 0.40] | Upgrading and improvement: take active measures to improve the performance of the project, focusing on indicators with a high demand for renewal, such as improving structural functions, maintaining ancillary facilities, enhancing the level of intelligence, improving the environment and landscape, and improving management |
| Orange interval | [0.40, 0.80] | Improvements and extensions: take active measures to improve the sustainability of the project and avoid deterioration, such as nodal modifications, section upgrades, lane extensions, construction of secondary roads, and application of new materials |
| Red interval | [0.80, 1.00] | Development and re-development: this refers to the overall pattern and design of the project to improve its unsustainable state, such as re-design and construction |

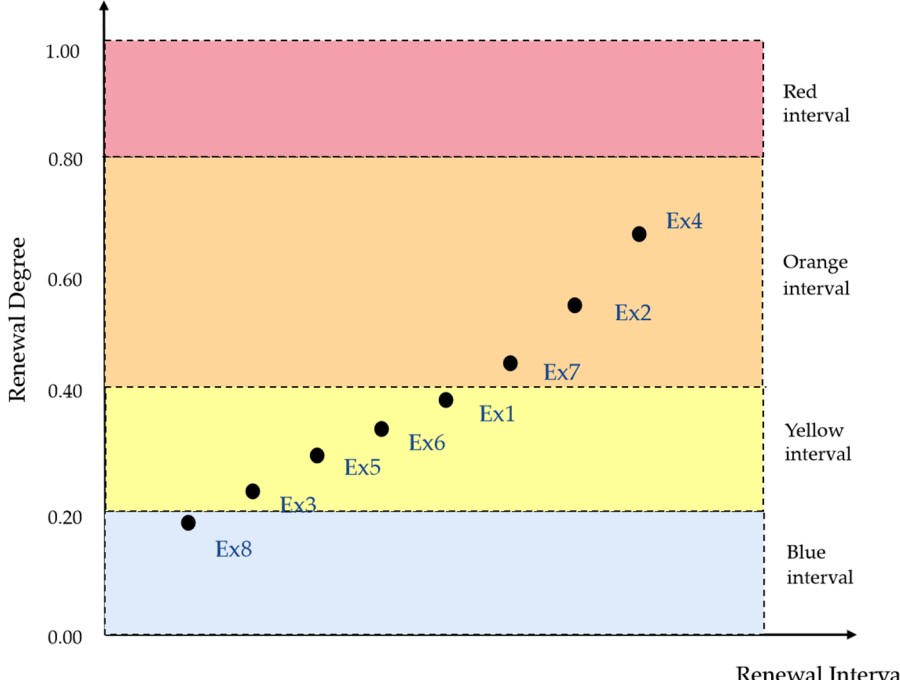

**Figure 3.** Expressway update interval map.

Based on the above description, we could divide the eight high and fast bars into intervals, where $C_i$ represents the update degree value, which shows that Ex4, Ex2, and Ex7 are in the orange update interval. Ex1, Ex6, Ex5, and Ex3 are in the yellow update interval, and Ex8 is in the blue update area. The schematic is shown in Figure 3.

Combining the above findings, we recommend a priority of Ex4 > Ex2 > Ex7 > Ex1 > Ex6 > Ex5 > Ex3 > Ex8 for these eight expressways. This priority is a comparative analysis that comprehensively places the expressways at the level of the citywide road network. The renewal intervals are assigned to reflect the urgency of renewing the expressways. According to the 2022 Shanghai Construction Plan, Ex4, Ex2, Ex7, and Ex1 were listed as projects that were to be updated and reconstructed, which verified the correctness of the results. Since the Ex8 project was renewed and transformed recently, its renewal degree value was low. The result was in line with the status quo of expressway renewal in Shanghai after evaluation by experts. When planning projects with limited funding, decision-makers can refer to the abovementioned priorities as a basis for decision-making.

## 5. Results and Discussion

### 5.1. Model Robustness Analysis

The value of $q$ in $q$-ROFSs reflects how optimistic the experts are about the values assigned [59]. $q$ was taken to be 3 in the above study. To verify the robustness of the model, we observed the impact of $q$ on the model results by adjusting the value of $q$ from 2 to 20 and then recalculating and ranking the expressway update values. As can be seen from Table 12, when the value is taken from 2 to 20, the results obtained by the model partially change. As observed in Figure 4, there is also a change in the update interval that the project is in. Referring to the interval values above, when $q$ is taken to be 15 and 20, the update interval for Ex2 changes to the yellow zone; when $q$ is taken to be above 5, the update interval for Ex7 and Ex8 also changes from the orange zone to the yellow zone and for Ex8 from the yellow zone to the blue zone. It can be seen that the evaluation model is stable for most of the updated intervals of the expressways under the change of the $q$ value, and the intervals of individual expressways change. This shows that the model is robust and effective.

**Table 12.** $C_i$ values for expressways at different q values.

| q | $C_i$ | | | | | | | |
|---|-----|-----|-----|-----|-----|-----|-----|-----|
|   | Ex1 | Ex2 | Ex3 | Ex4 | Ex5 | Ex6 | Ex7 | Ex8 |
| 2 | 0.389 | 0.485 | 0.302 | 0.624 | 0.351 | 0.374 | 0.419 | 0.234 |
| 3 | 0.388 | 0.480 | 0.300 | 0.628 | 0.341 | 0.364 | 0.413 | 0.198 |
| 4 | 0.384 | 0.471 | 0.296 | 0.638 | 0.328 | 0.348 | 0.401 | 0.189 |
| 5 | 0.379 | 0.460 | 0.290 | 0.648 | 0.313 | 0.331 | 0.389 | 0.180 |
| 6 | 0.373 | 0.450 | 0.285 | 0.658 | 0.300 | 0.314 | 0.378 | 0.161 |
| 7 | 0.368 | 0.440 | 0.281 | 0.667 | 0.287 | 0.299 | 0.368 | 0.143 |
| 8 | 0.364 | 0.432 | 0.278 | 0.675 | 0.277 | 0.285 | 0.360 | 0.126 |
| 9 | 0.359 | 0.424 | 0.276 | 0.682 | 0.267 | 0.273 | 0.353 | 0.111 |
| 10 | 0.355 | 0.418 | 0.275 | 0.687 | 0.259 | 0.263 | 0.347 | 0.097 |
| 15 | 0.342 | 0.398 | 0.274 | 0.701 | 0.236 | 0.231 | 0.333 | 0.050 |
| 20 | 0.336 | 0.390 | 0.275 | 0.706 | 0.227 | 0.218 | 0.330 | 0.031 |

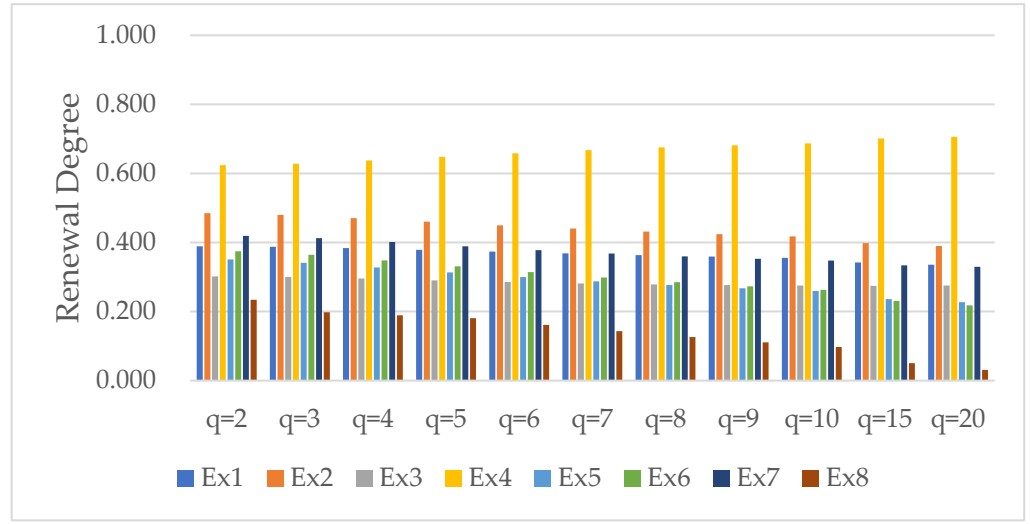

**Figure 4.** Interval of update degree values of expressways under different q values.

### 5.2. Suggestion for Decision-Making

To better understand the need for renewal of expressways, we took each expressway as an object and analyzed the main factors affecting its renewal degree value, which facilitates the development of a targeted renewal plan. Based on the matrix of $r_{ij}$ values obtained in step 2 of the TOPSIS method, it was possible to understand the score of each indicator of

the expressways and calculate it according to the $C_{Hi}$ scores of the secondary indicators included in the primary indicator ($H_i$) to obtain the score of the primary indicator ($H_i$) for each project. To make the results more intuitive, we represent them in a pie chart, as shown in Figure 5.

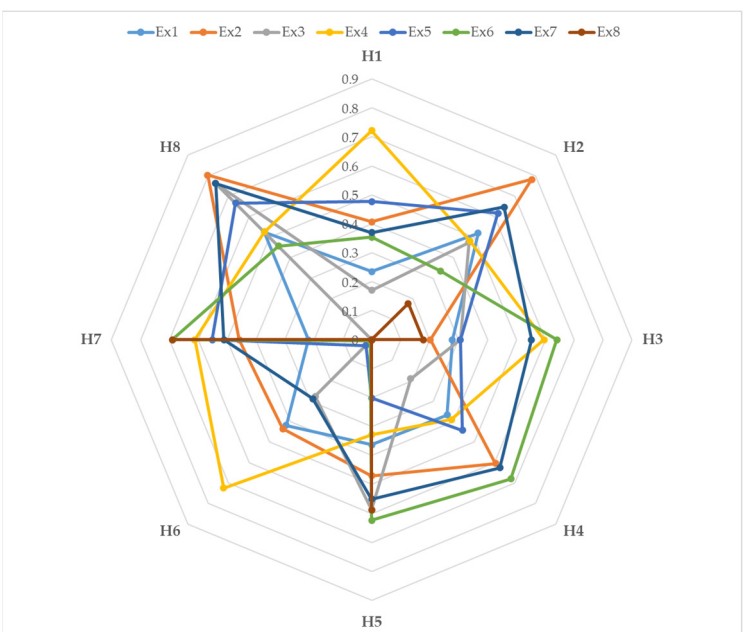

**Figure 5.** Graph of factors influencing the renewal of each expressway.

In the above chart, the $H_i$ value indicates the value of the degree of influence of the dimension indicator on the renewal degree of the expressway; the higher the value, the higher the renewal demand of this evaluation dimension.

(1) For Ex4 (the G15 Expressway—Jia Liu section), the $C_{Hi}$ values for structural renewal demand (H1) and traffic development demand (H6) exceed 0.7, and the higher $C_{Hi}$ values for natural resilience demand (H3) and social development demand (H7) exceed 0.6. The bridge structure of this expressway is deteriorating, the maintenance work is on a significant upward trend, and the service level is rated as level 4. Moreover, this expressway belongs to the north–south access route between Shanghai and Jiangsu Province, and there are more traffic accidents every year; therefore, the project should develop a targeted renewal strategy in terms of upgrading the structural function of the facilities, expanding the traffic capacity, and improving the safety performance.

(2) For Ex2 (the G50 Shanghai–Chongqing Expressway), factors such as service function renewal demand (H2), environmental development demand (H8), and traffic development demand (H6) have a greater impact on the renewal degree value, with $C_{Hi}$ values exceeding 0.7. As the project is located in an area where large factories and industrial parks are located, the restructuring of the industrial layout brings a greater impact on traffic demand and frequent traffic congestion during weekends and festivals. Coupled with problems such as low public satisfaction, these issues should encourage the project to develop corresponding renewal strategies in terms of adapting to traffic growth, improving road service levels, performing green construction, and reducing carbon emissions.

(3) For Ex7 (the Hu Song Road Jia Min Elevated—S32), for environmental development needs (H8), $C_{Hi}$ values reach 0.7 or more. Service function renewal demand (H2) and institutional resilience (H4) have a greater impact on the renewal degree value, with $C_{Hi}$ values reaching 0.6 or more. The project has been under construction for a long time, and as the only arterial link between the central city and the south and western parts of Shanghai, it faces heavy traffic congestion, while damaging the city's environmental image. The average length of time of emergency response and handling after a major disaster exceeds the length of time specified by the management; therefore, its management

coordination and incident handling capabilities should be improved. Furthermore, the expressway should be renewed in terms of improving environmental pollution, enhancing the appearance of the project, and reducing carbon emissions.

(4) For Ex1 (the S20 West Section of the Outer Ring Road), the $C_{Hi}$ values of environmental development demand (H8) and service function renewal demand (H2) reaching 0.5 or more. Owing to the frequent congestion on the road section during peak hours and the upgrading of the Hongqiao area and Minhang area where the project is located, the traffic function demand is increasing. Coupled with the more serious environmental pollution in this area, these issues make it necessary to develop corresponding renewal strategies in terms of upgrading the service level of facilities, performing green construction, and committing to energy saving and emission reduction.

(5) For Ex6 (the Yanggao Road, Zhouhai Road–Jinhai Road), social development demand (H7), economic development demand (H5), institutional resilience enhancement demand (H4), and natural resilience enhancement demand (H3) are more influential, with $C_{Hi}$ values reaching 0.6 or more. The project is an important north–south traffic road in Pudong New Area, but as its construction age has reached 30 years, its normal traffic has been affected by natural disasters, especially heavy rainfall disasters, more often than not every year. Thus, its renewal strategy should pay special attention to drainage, permeability, and other related designs. In the event of a major disaster, the average length of time of emergency response and treatment exceeds the time set by the relevant authorities; therefore, the management's ability to coordinate and deal with the incident should be improved. Moreover, the rapid economic development of the area in which it is located requires it to be updated as soon as possible to meet the needs of the social environment.

(6) For the Ex5 project (the G15 Expressway—Jiajin section), the $C_{Hi}$ values for service function renewal needs (H2), social development needs (H7), and environmental development needs (H8) reach 0.6 or more. Owing to its high saturation, low traffic fluency, and medium level of knowledge, there is an urgent need for improvement in service functions. In addition, the high rate of population growth and the large proportion of the working population in this area make its social renewal needs high. In terms of environmental development needs, the air pollution index is high; therefore, a corresponding regeneration strategy is needed to improve environmental issues.

(7) For Ex3 (the Inner Ring Elevated Siping Road—Zhengben Road), the environmental development demand (H8) is more urgent, with a $C_{Hi}$ value of 0.7 or more, and the economic development demand (H5) is higher, with a $C_{Hi}$ value of 0.6 or more. As the Inner Ring Elevated Siping Road is in the city center and has a service life of 26 years, its environmental pollution problems are greater, especially noise pollution, for which corresponding improvement strategies should be developed. The project also needs to develop a targeted renewal strategy in terms of its harmony with the urban environment. Moreover, the development of the economic level of the area in which it is located requires the comprehensive upgrading of its service functions through renewal.

(8) For Ex8 (the Jiyang Road Lupu Bridge—Minhang District boundary), this expressway is in the blue interval in terms of the renewal degree values. Since it was renewed in 2021, the renewal demand is relatively low. However, in terms of individual dimensions, its social renewal demand and economic renewal demand values are also relatively high, and more attention should be paid to the impact of these two aspects in a follow-up.

Combined with the results discussed in Section 4.2, these defined renewal intervals can provide decision-makers with a clearer picture of the state of renewal demands of an expressway. When an expressway is in the red renewal zone, it is in a state of serious imbalance and has an urgent need for renewal and it is recommended to carry out development and reconstruction, such as comprehensively optimizing the design, changing its original construction form, and improving the overall layout. Conversely, when an expressway is in the blue renewal zone, it is in a relatively sustainable state for the time being and can continue to be maintained in its current state. Furthermore, it is recommended to carry out repairs and maintenance, pay close attention to changes in the updated demand indicators,

and maintain its healthy operation, as shown in Ex8. The roads in the orange area can be improved through reconstruction and expansion to avoid deterioration of conditions, such as node reconstruction, road section improvement, lane expansion, auxiliary road construction, and the application of new materials. It should be noted that the roads in the orange zone, such as Ex2 and Ex7, do not differ much in their renewal degree values, but the former has a higher demand for service functions and traffic development, while the latter has a more prominent demand for environmental development. Therefore, in the decision-making process, the renewal priority can be considered per the general development strategy objectives of the city. For example, if a city sets emission reduction and carbon peaking as its strategic objectives in subsequent years, it can prioritize the renewal needs of Ex7. For roads in the yellow interval, measures should be taken to upgrade and improve project performance, such as improving structural functions, maintaining auxiliary facilities, improving the level of intelligence, improving the environment and landscape, and improving management levels, such as Ex1, Ex6, Ex5, and Ex3. The above suggestions can help decision-makers make more informed decisions on expressway renewal. In addition, the evaluation of the renewal degree of expressways plays a positive role in the rational allocation of road resources, optimization of road grades, effective planning of land use, and improvement of the overall style of the city in the process of urban future development planning.

## 6. Conclusions and Future Research

The evaluation of the degree of renewal of urban expressways is a necessary and important part of decision-making. This study improves the application of the concept of urban renewal in the field development of urban expressways and establishes an evaluation indicator system and decision-making framework that combines qualitative and quantitative evaluations, which are conducive to the sustainable renewal of urban expressways. There are some contributions in our work. First, a hybrid evaluation framework was established using the FDM-FAHP method and the TOPSIS method. In addition, the FDM based on q-ROFSs which is a powerful and effective tool to describe uncertainty and vagueness, which was developed to overcome the inherent restrictions of the traditional Delphi method. The FAHP based on q-ROFSs was adopted for computing the weights of indicators on the assessment matrix. The q-ROFSs was utilized to substitute classical fuzzy sets and their extensions, which has a good performance in the sensitivity analysis. TOPSIS was presented and utilized to rank the alternative expressways and calculate the renewal degree. Secondly, the entire set of regeneration indicators was constructed and considered from macro, meso, and micro perspectives, and the five factors with the greatest impact on expressway renewal were the demand for transport development, the renewal of facility and service functions, the upgrading of institutional resilience, structural renewal, and economic development. The results of the assessment fully reflected the renewal demand of the expressways to adapt to urban development. Finally, the model revealed the renewal degree and the strengths and weaknesses of each expressway. Thus, it is well suitable for decision-makers who need to choose priority development strategies for expressways based on their characteristics and improve the rationality of the practical decision-making process.

The study still has a few limitations and therefore offers some future research directions. First, urban expressway renewal is a new topic that promotes the sustainable renewal of urban roads. In the future, our proposed framework can be modified by adding more indicators that are responsible for UERD. Second, researchers can extend this study by developing mathematical models. Different fuzzy language sets can be compared to analyze their stability. During the analysis of the expressway ranking, the limitation of the TOPSIS method is easily affected by the evaluation scheme set. Some methods such as ELECTRE Tri [60] can be used for comparative analysis. In addition, expressway renewal values may change over time; therefore, a dynamic model is recommended for further comparison and for sustainable development.

**Author Contributions:** All authors contributed to the conceptualization of this study, as well as the methodology and data analysis. Conceptualization: M.Z., W.Z. and M.H.; data curation: M.Z. and J.D.; formal analysis: M.Z.; funding acquisition: W.Z.; investigation: M.Z. and J.D.; and methodology: M.Z. and M.H. project administration: M.H. and J.D.; resources: W.Z. and J.D.; software: M.Z.; supervision: W.Z., M.H. and J.D.; validation: T.Y.; writing—original draft: M.Z.; and writing—review & editing: M.H. and T.Y. All authors have read and agreed to the published version of the manuscript.

**Funding:** This work was supported by the Science and Technology Commission of Shanghai Municipality Project (20DZ2251900 and 21ZR1423800); Humanities and Social Science Research Project of colleges and universities in Jiangxi Province (GL21219).

**Institutional Review Board Statement:** Not applicable.

**Informed Consent Statement:** Not applicable.

**Data Availability Statement:** The data presented in this study are available on request from the corresponding author. The data are not publicly available because they belong to the road operation management companies.

**Conflicts of Interest:** The authors declare no conflict of interest.

## Appendix A

**Table A1.** Expressway Information Sheet.

| Expressway | Ex1 | Ex2 | Ex3 | Ex4 | Ex5 | Ex6 | Ex7 | Ex8 |
|---|---|---|---|---|---|---|---|---|
| Area of affiliation | Hongkou District, Minhang District | Qingpu District, Minhang District | Yangpu District | Jiading District | Jiading District, Jinshan District | Pudong New District | Songjiang District, Jinshan District | Pudong New District |
| Status of facilities | -Built in 1997, opened to traffic in 2003, with eight lanes in both directions -From Tao Pu Road in Putuo District in the north to Xin Zhu Road in Minhang District in the south, eight lanes in both directions, for motor vehicles only, with a design speed of 80 km/h. The total length is approximately 18 km. | -Twenty years of service life, with no major repairs performed during the operational period -Four-lane roads in both directions (approximately 26.5 km), with a current one-way flow of 6000 pcu/h, causing heavy traffic congestion during peak periods -Pavement structural strength index (PSSI) "poor" rate of nearly 85%, making the road a class IV service level -Poor state of service of facilities -Congested roads during peak periods -Improve the east–west trunk of the high-speed network to achieve rapid regional interchange -Industry transformation and upgrading along the route: Huawei Qingpu Base, City West Software Park, increasing demand for transportation functions -Low level of digitization | -Opened in 1994, with a total length 47.7 km -High operating turnover, slow overall operating speed, and a full-day traffic turnover of 2.71 million pcu vehicles/km -Technical condition assessment category B -Noise-environmental problems, outdated landscaping, and broken crash walls, resulting in incongruity with the urban landscape | -Built in 2001, with a design speed of 100 km/h and six lanes in both directions -Maintenance work is on the rise -The structural deterioration of the bridge is accelerating -Traffic volume continues to grow -The road is at a class IV level of service | -North to the northern section of the Shanghai Bypass Expressway (G1503) and south to the Shenjiahu Expressway (S32), approximately 44 km long -One of the main north–south national highway routes | -Built in 1992, the road is 6.6 km long and has a planned red line width of 50 m -An important north–south corridor in Pudong New Area | -The total length of the road is approximately 22.3 km, starting from Jiamin Elevated Road in the north and ending at Shenjiahu Expressway (S32) in the south -The road grade is secondary road, with a calculated speed of 60 km/h -Traffic congestion, poor surrounding environment, and unreasonable industrial structure along the route | -Approximately 7.1 km from the approach bridge of the Lupu Bridge in the north to the Minhang District boundary in the south -Main north–south access road in the area |
| Renewal requirements | -Congestion needs to be eased on high traffic peak roads -The Hongqiao area and Minhang area are being upgraded to become the main city area with increasing traffic demand | | -Improve the safety of the facility itself -Improve the urban landscape and adapt to future urban development and build model roads for urban road renewal -The need for new infrastructure and intelligent development -Severe noise and environmental pollution and traffic congestion -Improve citizen satisfaction | -Congestion needs to be relieved and upgraded to level 3 service standards -Non-prestressed bridges need to be strengthened -Difficult to maintain on a daily basis and more safety issues at night -Need to connect with Jiangsu Province -Large residential communities are planned along the route in the northern part of Jiading Industrial Zone | -For accelerating the construction of Hongqiao International Open Hub and building a new urban area -Promote the reconstruction of the traffic bottleneck section | -Old road surface -Congestion needs to be eased on high traffic peak roads | -The only arterial transport link between the central city and the southwest of Shanghai -To connect Songjiang hub and Hongqiao hub -Need to divert traffic flow from G60 into the city | -Meeting the needs of regional population growth -Meeting the needs of rapid regional economic development |

## Appendix B

**Table A2.** The Distance between PIS and NIS for Each Expressway.

| Indicators/Expresssway | $d^+$ | | | | | | | | $d^-$ | | | | | | | |
|---|---|---|---|---|---|---|---|---|---|---|---|---|---|---|---|---|
| | Ex1 | Ex2 | Ex3 | Ex4 | Ex5 | Ex6 | Ex7 | Ex8 | Ex1 | Ex2 | Ex3 | Ex4 | Ex5 | Ex6 | Ex7 | Ex8 |
| Hardware facilities C1 | 0.005 | 0.005 | 0.005 | 0.000 | 0.001 | 0.005 | 0.005 | 0.008 | 0.003 | 0.003 | 0.003 | 0.008 | 0.007 | 0.003 | 0.002 | 0.000 |
| Facilities maintenance C2 | 0.004 | 0.001 | 0.006 | 0.005 | 0.004 | 0.000 | 0.000 | 0.006 | 0.002 | 0.005 | 0.000 | 0.001 | 0.002 | 0.006 | 0.006 | 0.000 |
| Driving safety C3 | 0.011 | 0.009 | 0.011 | 0.000 | 0.008 | 0.012 | 0.010 | 0.012 | 0.001 | 0.003 | 0.001 | 0.012 | 0.004 | 0.000 | 0.002 | 0.000 |
| Road saturation C4 | 0.003 | 0.001 | 0.003 | 0.005 | 0.000 | 0.006 | 0.003 | 0.004 | 0.003 | 0.005 | 0.002 | 0.000 | 0.006 | 0.000 | 0.003 | 0.002 |
| Traffic accessibility C5 | 0.001 | 0.000 | 0.000 | 0.001 | 0.003 | 0.001 | 0.000 | 0.003 | 0.002 | 0.003 | 0.003 | 0.003 | 0.000 | 0.002 | 0.003 | 0.000 |
| Level of intelligent control C6 | 0.004 | 0.002 | 0.005 | 0.002 | 0.002 | 0.003 | 0.000 | 0.005 | 0.001 | 0.003 | 0.000 | 0.003 | 0.002 | 0.002 | 0.005 | 0.000 |
| Public satisfaction C7 | 0.002 | 0.000 | 0.001 | 0.001 | 0.003 | 0.001 | 0.001 | 0.004 | 0.002 | 0.004 | 0.003 | 0.004 | 0.002 | 0.003 | 0.004 | 0.000 |
| Road Importance C8 | 0.001 | 0.001 | 0.002 | 0.004 | 0.000 | 0.005 | 0.003 | 0.005 | 0.004 | 0.004 | 0.003 | 0.002 | 0.005 | 0.000 | 0.002 | 0.000 |
| Storm resilience C9 | 0.004 | 0.003 | 0.003 | 0.001 | 0.003 | 0.000 | 0.001 | 0.004 | 0.000 | 0.000 | 0.001 | 0.003 | 0.001 | 0.004 | 0.002 | 0.000 |
| Earthquake resilience C10 | 0.000 | 0.002 | 0.000 | 0.002 | 0.000 | 0.002 | 0.002 | 0.002 | 0.002 | 0.000 | 0.002 | 0.000 | 0.002 | 0.000 | 0.000 | 0.000 |
| Landslide resilience C11 | 0.001 | 0.000 | 0.001 | 0.000 | 0.001 | 0.000 | 0.001 | 0.001 | 0.000 | 0.001 | 0.000 | 0.001 | 0.000 | 0.001 | 0.000 | 0.000 |
| Road surface ice resilience C12 | 0.001 | 0.002 | 0.001 | 0.002 | 0.001 | 0.002 | 0.000 | 0.001 | 0.001 | 0.000 | 0.001 | 0.000 | 0.001 | 0.000 | 0.002 | 0.001 |
| Fire resilience C13 | 0.003 | 0.003 | 0.003 | 0.000 | 0.003 | 0.000 | 0.002 | 0.003 | 0.000 | 0.000 | 0.000 | 0.003 | 0.000 | 0.003 | 0.002 | 0.000 |
| Efficiency of emergency response to major disasters C14 | 0.010 | 0.007 | 0.009 | 0.005 | 0.007 | 0.000 | 0.003 | 0.010 | 0.001 | 0.004 | 0.002 | 0.006 | 0.004 | 0.010 | 0.008 | 0.000 |
| Energy conservation management system C15 | 0.000 | 0.001 | 0.001 | 0.000 | 0.001 | 0.003 | 0.001 | 0.004 | 0.004 | 0.003 | 0.003 | 0.004 | 0.003 | 0.001 | 0.003 | 0.000 |
| Accessibility of information C16 | 0.005 | 0.000 | 0.009 | 0.009 | 0.005 | 0.005 | 0.005 | 0.009 | 0.005 | 0.009 | 0.000 | 0.000 | 0.005 | 0.005 | 0.005 | 0.000 |
| Policy direction C17 | 0.002 | 0.000 | 0.002 | 0.002 | 0.003 | 0.003 | 0.005 | 0.002 | 0.003 | 0.005 | 0.003 | 0.003 | 0.002 | 0.002 | 0.000 | 0.003 |
| Gross regional domestic product C18 | 0.003 | 0.003 | 0.004 | 0.003 | 0.003 | 0.000 | 0.001 | 0.003 | 0.001 | 0.002 | 0.000 | 0.001 | 0.001 | 0.004 | 0.003 | 0.001 |
| Regional infrastructure investment C19 | 0.005 | 0.005 | 0.000 | 0.005 | 0.005 | 0.002 | 0.001 | 0.002 | 0.001 | 0.000 | 0.005 | 0.000 | 0.000 | 0.004 | 0.005 | 0.004 |
| Cross-regional traffic flow C20 | 0.023 | 0.021 | 0.023 | 0.000 | 0.022 | 0.023 | 0.022 | 0.023 | 0.000 | 0.002 | 0.000 | 0.023 | 0.001 | 0.000 | 0.001 | 0.000 |
| Industrial layout requirements C21 | 0.000 | 0.000 | 0.013 | 0.000 | 0.013 | 0.013 | 0.013 | 0.013 | 0.013 | 0.013 | 0.013 | 0.000 | 0.013 | 0.000 | 0.000 | 0.000 |
| Characteristic development needs C22 | 0.000 | 0.000 | 0.000 | 0.010 | 0.010 | 0.010 | 0.000 | 0.010 | 0.010 | 0.010 | 0.010 | 0.000 | 0.000 | 0.000 | 0.010 | 0.000 |
| Population growth C23 | 0.007 | 0.004 | 0.010 | 0.000 | 0.002 | 0.004 | 0.003 | 0.004 | 0.003 | 0.006 | 0.000 | 0.010 | 0.008 | 0.006 | 0.007 | 0.006 |
| Demographic structure C24 | 0.000 | 0.000 | 0.001 | 0.000 | 0.000 | 0.000 | 0.000 | 0.000 | 0.000 | 0.001 | 0.000 | 0.001 | 0.000 | 0.000 | 0.000 | 0.000 |
| Employment position C25 | 0.007 | 0.007 | 0.007 | 0.006 | 0.007 | 0.000 | 0.007 | 0.000 | 0.000 | 0.000 | 0.000 | 0.000 | 0.000 | 0.007 | 0.000 | 0.007 |
| Environmental pollution assessment C26 | 0.000 | 0.000 | 0.000 | 0.000 | 0.000 | 0.000 | 0.000 | 0.000 | 0.000 | 0.000 | 0.000 | 0.000 | 0.000 | 0.000 | 0.000 | 0.000 |
| Carbon emission assessment C27 | 0.002 | 0.000 | 0.002 | 0.002 | 0.002 | 0.006 | 0.002 | 0.006 | 0.004 | 0.006 | 0.004 | 0.004 | 0.004 | 0.000 | 0.004 | 0.000 |
| Harmony with the urban environment C28 | 0.003 | 0.002 | 0.000 | 0.003 | 0.002 | 0.000 | 0.000 | 0.005 | 0.002 | 0.003 | 0.005 | 0.002 | 0.003 | 0.005 | 0.005 | 0.000 |

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
