# Peer review of "Evaluating the Renewal Degree for Expressway Regeneration Projects Based on a Model Integrating the Fuzzy Delphi Method, the Fuzzy AHP Method, and the TOPSIS Method"

_sustainability, doi:10.3390/su15043769_

Round 1

Reviewer 1 Report

This paper systematically proposed a generic and quantifiable indicator system for evaluating the degree of renewal of urban expressways. The author's introduction to the research background and methods for evaluation are complete enough. However, there are still some problems, which must be solved before it is considered for publication.

(1) The format of the tables in the article is not standardized. For example, Tables 7 and 9 are missing headers when spanning pages.

(2) When collecting data for the experiment, only the data of Shanghai 2021 was collected. The year is too single and the amount of data is too small, which is easily affected by chance factors. It is recommended to add more years of data for the experiment.

(3) In the Conclusions section, future research directions can be appropriately mentioned so as to provide further research ideas for subsequent researchers.

(4) The references can be improved by adding some new and closed related on MCDM, such as https://doi.org/10.1007/s10726-021-09735-0, http://doi.org/10.1111/exsy.12681;.

In my opinion, this paper is interesting and valuable, but a review is needed before it can be reconsidered for publication.

Author Response

Dear Reviewers,

Thank you for your kind considerations and comments on our manuscript entitled “Evaluating the Renewal Degree for Expressway Regeneration Projects Based on a Model Integrating Fuzzy Delphi Method-Fuzzy AHP and TOPSIS” (Manuscript ID: sustainability-2159619). It is quite helpful for improving the quality of our paper and we have studied comments carefully and have made revision and explanation according to your comments.

The revisions have been marked up using the “Track Changes” function in Microsoft Word.

The main corrections in the paper and the responses to the reviewers' comments point by point are in the attachment.

We appreciate for respected reviewers’ warm work earnestly, and hope that the correction will meet with approval. We look forward to hearing from you.

Best regards,

Min Zhu

Wenbo Zhou

HU-UTS SILC Business School, Shanghai University,

Shanghai 201800, China 

E-mail address: zhumin68@shu.edu.cn

Reviewer 2 Report

Review to “Sustainability-2159619” manuscript

1. Positive aspects

“Sustainability-2159619” manuscript is located as topic in the area of ​​interest of „Sustainability”journal.

This manuscript has an actual and interesting research idea. The abstract presents objects, methods and results. Following the calculations, they appear to be correct in mathematical order. This manuscript has substantial, special merits in terms of practical impact.

2. Negative aspects

Given that

a) the current research focuses on problems of one of the largest cities on the planet in terms of population and number of skyscrapers (5th place after (Hong Kong, New York, Shenzen and Dubai)) and

   b) the current research does not have a “Future research” section,

we believe that it is useful to consult a recent study on urban renewal in the city of Lahore (Pakistan), both because Lahore is also one of the great and old cities of the world (although the study in question focuses on the activities of “renewal sites”) (Ali, R., Waqar, Z., Ahmad, M., & Sharif, M. R. (2022). Assessing Urban Renewal Sites in Lahore. Journal of World Science, 1(11), 1038-1047), as well as to possibly complete the system of indicators used and/or the “five factors with the greatest impact on renewal”.

3. Conclusion:

„Sustainability-2159619” manuscript requires minor completion and addition:

a) addition of “Future research” section (especially since the well-developed manuscript refers to “future development” and “future discovery”);

b) completing the bibliography.

Author Response

Dear Reviewer,

Thank you for your kind considerations and comments on our manuscript entitled “Evaluating the Renewal Degree for Expressway Regeneration Projects Based on a Model Integrating Fuzzy Delphi Method-Fuzzy AHP and TOPSIS” (Manuscript ID: sustainability-2159619). It is quite helpful for improving the quality of our paper and we have studied comments carefully and have made revision and explanation according to your comments.

The revisions have been marked up using the “Track Changes” function in Microsoft Word.

The main corrections in the paper and the responses to the reviewers' comments point by point are in the attachment.

We appreciate for respected reviewers’ warm work earnestly, and hope that the correction will meet with approval. We look forward to hearing from you.

Best regards,

Min Zhu

Wenbo Zhou

HU-UTS SILC Business School, Shanghai University,

Shanghai 201800, China 

E-mail address: zhumin68@shu.edu.cn

Reviewer 3 Report

Thank you for giving me the opportunity to read your paper. The paper “Evaluating the Renewal Degree for Expressway Regeneration Projects Based on a Model Integrating FDM-FAHP and TOPSIS” is interesting for journal readers. But following changes should be done before the consideration to improve the quality of the paper:

Author's name and their addresses should not be appeared in manuscript as this is against the prescribed journal instructions. This manuscript failed to present the study debates and failed to discuss the debates. The readability is low and the scientific contribution is short discussed.

Abstract: Abstract is incomplete. I suggest authors to rewrite the abstract to make it more constructive with clear methodology, clear results and findings. Abstract should have at least one sentence per each: context and background, motivation, hypothesis, methods, results, conclusions. Novelty is missing in abstract.

Introduction: The introduction should include problem context, literature review and the research gap analysis of the previously published research. Research questions and research objectives are missing. Authors are recommended to spend much time revising this section.

LR: The LR can be written more attractive; need improvements to cite more latest literature, also give more the touch of expressway regeneration, renewal degree, fuzzy delphi method, fuzzy AHP, q-rung orthopair fuzzy sets, TOPSIS. Further, in the last paragraph need to more clearly mention that what's novel in this study and how it is going to contribute in the existing literature. Many similar works have been published already so no major novelty here. Here are some studies to cite in the introduction as well as in the literature 2. The literature review should be more up to date.

RM: How did the authors get from the theoretical model to the empirical one? Behind the model there need to be a complete and well-thought-out theoretical grounding. This part of the article shouldn't include any citations or references; rather, it should be structured according to the authors' reasoning. The empirical model will come when this part has been completed. Write demographic details of all respondents.

Data analysis and Results: The authors have only presented the findings, with no explanation of their economic reasoning. Do these findings validate or disprove the current policy framework? Are any new policy measures planned as a result of the findings? Discussion of the findings, which is conspicuously absent here, is meant to spark debate on policy. If the results don't offer anything new in terms of theory or policy, then a simple comparison with the literature won't prove their originality.

Discussion: Discussion is short in the text, expand it more. And compare with similar case and papers. There are theoretical, managerial and societal implications missing.

It would be appropriate to indicate future research directions and limitations of this at the end of the conclusion section just before references. Need clear future recommendation/implementation in the context of patent citations and knowledge flow.

Improve the language of the whole manuscript.

Poor references format: authors should address this issue in the revised draft. Kindly follow the right style of citation (references) throughout the manuscript by checking the guidelines of journal or any previously published paper in the journal.

Hence, I would like to recommend this manuscript major revision.

Author Response

(The authors gave the same response as above.)

Reviewer 4 Report

I have no major comments to the reviewed work. Literature, description of figures and tables should be improved. You should also check the correctness of the language. In addition, the article may be published in a journal.

Author Response

(The authors gave the same response as above.)

Reviewer 5 Report

See attached

Author Response

(The authors gave the same response as above.)

Reviewer 6 Report

The article presents an interesting and relevant topic of research; however, as I will explain in my review, I don’t think the method used is the most appropriate one for what the authors intend to.

- Reference list needs articles DOI

LINE 31: “Urban expressways play a vital role in connecting and driving the development of cities as transport hubs and backbones, and their regeneration has received increasing attention” - This needs references.

LINE 33: “The volume of urban expressways in major cities around the world is high.” - I’m not sure about the need for this phrase here. I understand why it’s written, it could use a little bit more connection to the text. Additionally, maybe a reference would be adequate here.

LINE 34: “Its transport facilities investment amounted to RMB 947.51 billion between 2003 and 2017,  and relevant industry data show that the total length of Shanghai's road network reached 18,500 km in 2022, with an average operating life of 12.5 years.” - Needs references

LINE 37: “Thus, roads now face the problem of old age.” - Too colloquial; rewrite in a more proper manner.

LINE 42: “Urban high expressway renewal is the main way to make up for such planning drawbacks, enhance functions and services, improve environmental impacts, and increase the convenience of travel for citizens.” - It's a little bold to say that is the main way without a reference. I would change main way to something like “one of the main ways” and find some references to corroborate it.

LINE 84: “To improve the road assessment system, scholars in China and abroad have conducted much research.” - First, it needs references, there might be already in use in the article but it needs them anyway. Second I would change the phrase. Something like: In the past decades, both abroad and in China, a large body of research has been published regarding the improvement of road assessment systems…”. Just rewrite the phrase to be more polished.

LINES 88-103: In all the examples the plural is used: “some studies” / “other studies” but there is only one reference. Please add more. While the reference [10] is a literature review article the remaining aren’t. Thus, when giving examples of what was made, refer to more than one article. I would suggest 3 for each as a good amount.

LINE 102: “and most were limited to qualitative evaluation, with indicators lacking specific quantitative methods.” Which ones? Reference them.

LINE 104-120: The same as above. Everything is in the plural but only one reference is mentioned. More references should be added.

3.5. TOPSIS method for the order of preference by similarity to ideal solution

LINE 409: “TOPSIS is one of the most commonly used methods for measuring the distance between evaluation solutions and ideal solutions.” - There is no question that TOPSIS is one of the most commonly used method in multicriteria analysis and in many practical applications of engineering but “measuring the distance between evaluation solutions and ideal solutions” is not why the method is picked, rather a characteristic of the method.

“TOPSIS is one of the most commonly used methods for measuring the distance be”tween evaluation solutions and ideal solutions.” - Again, true but it also has some disadvantages and that’s why in some cases using ELECTRE I, III, TRI or PROMETHEE (just to mention some that are not compensatory) sometimes are better choices. If an alternative is not considered, TOPSIS needs to be rerun again and the results might be different. In this specific case, if an expressway is taken out of the equation, TOPSIS needs to be re-run for the remaining ones.
It’s not mentioned in the article its disadvantages and if it can (or not) add some limitations to the research. At some point would be ideal addressing these issues.

Although already in the subsection tittle, TOPSIS should be at least once written in full.

It measures the distance to the best solution (ideal) but also to the anti-ideal, a point that should also be mentioned.

When normalising to use TOPSIS, you can pick between different normalisations (vectorial, linear,…) It would be ideal to explain how and why did you pick the normalisation used.

LINE 476: “Next, the assessment status was determined. According to the warning levels corresponding to different colors in the disaster forecast and the standard division of the correlation degree [48], the degree of urgency of expressway renewal was divided into four intervals (i.e., the blue renewal interval, yellow renewal interval, orange renewal interval, and red renewal interval), and their score intervals are shown in Table 11. According to the grade division of the interval value where the target value was located, the renewal urgency of the expressway and the renewal status result were obtained (Figure 3).”

Now this changes the picture completely. So, TOPSIS is being used but it’s also being created different levels to assess the urgency of renewal within the TOPSIS results? This is just ELECTRE Tri territory with all the negative points of TOPSIS. Why was TOPSIS chosen instead of ELECTRE Tri?
Unlike ELECTRE I, III or PROMETHEE, ELECTRE Tri is not hard to understand,.It is more complex to create but it and can be easily understood by municipal authorities and the public. And the best part is that it creates the levels from the get-go and its non-compensatory, meaning that there is no need to repeat the method when an expressway is not considered.

Assigning alternatives (like expressways) to the most appropriate class according to their overall performance is a multicriteria problem of the ‘sort problematic’ kind. ELECTRE Tri is one of the most used non-compensatory methods for this purpose because it does the assignment in a way that mimics human judgement. Each class is delimited by upper and lower profiles, or ‘reference alternatives’, for each criterion, whose values may be defined by codes of practice or decision-maker choice.
The method compares criteria values of the alternatives against values of the reference alternatives, utilising an outranking procedure to assign ultimately a class to the alternatives. It considers indifference, preference and veto thresholds to accommodate in a natural way the imprecision inherent to human decision processes. Applying ELECTRE Tri requires defining the aforementioned thresholds, criteria weights and a cut-off parameter, l, and a class assignment rule (pessimistic or optimistic).

My recommendation is to seriously think about changing TOPSIS to ELECTRE Tri. TOPSIS provides a rank while ELECTRE Tri a classification system, which is in fact what has been made in the article. ELECTRE Tri will provide the results intended with much more confidence and in much more detail. I will leave the main bibliography regarding ELECTRE Tri and how the method work below. There are also plenty of cases and applications to transportation and urban planning of the method.

Most important ELECTRE Tri bibliography:

Figueira J, Mousseau V and Roy B (2005) ELECTRE methods. In Multiple Criteria Decision Analysis: State of the Art Surveys (Figueira J, Greco S and Ehrogott M (eds)). Springer, New York, NY, USA, pp. 133–153.

Mousseau V, Slowinski R and Zielniewicz P (2000) A user-oriented implementation of the ELECTRE-TRI method integrating preference elicitation support. Computers and Operations Research 27(7–8): 757–777, https://doi.org/10.1016/S0305-0548(99)00117-3.

Mousseau V, Figueira J and Naux JP (2001) Using assignment examples to infer weights for ELECTRE TRI method: some experimental results. European Journal of Operational Research 130(2): 263–275, https://doi.org/10.1016/S0377-2217(00)00041-2.

Author Response

(The authors gave the same response as above.)

Round 2

Reviewer 1 Report

This paper have revised this paper carefully according to my suggestions and comments, and can be considered for publication.

Reviewer 3 Report

revised substantially.

Reviewer 6 Report

No more comments to add